# Imagine That!
# Leveraging Emergent Affordances for 3D Tool Synthesis

## Abstract

In this paper we explore the richness of information captured by the latent space of a vision-based generative model. The model combines unsupervised generative learning with a task-based performance predictor to learn and to exploit task-relevant object *affordances* given visual observations from a reaching task, involving a scenario and a stick-like tool. While the learned embedding of the generative model captures factors of variation in 3D tool geometry (e.g. length, width, and shape), the performance predictor identifies sub-manifolds of the embedding that correlate with task success. Within a variety of scenarios, we demonstrate that traversing the latent space via backpropagation from the performance predictor allows us to *imagine* tools appropriate for the task at hand. Our results indicate that affordances – like the utility for reaching – are encoded along smooth trajectories in latent space. Accessing these emergent affordances by considering only *high-level* performance criteria (such as task success) enables an agent to manipulate tool geometries in a targeted and deliberate way.

## 1 Introduction

The advent of deep generative models (e.g. Burgess et al., 2019; Greff et al., 2019; Engelcke et al., 2019) with their aptitude for unsupervised representation learning casts a new light on learning *affordances* (Gibson, 1977). This kind of representation learning raises a tantalising question: Given that generative models naturally capture factors of variation, could they also be used to expose these factors such that they can be modified in a task-driven way? We posit that a task-driven traversal of a structured latent space leads to *affordances* emerging naturally along trajectories in this space. This is in stark contrast to more common approaches to affordance learning where it is achieved via direct supervision or implicitly via imitation (e.g. Tikhanoff et al., 2013; Myers et al., 2015; Liu et al., 2018; Grabner et al., 2011; Do et al., 2018). The setting we choose for our investigation is that of tool synthesis for reaching tasks as commonly investigated in the cognitive sciences (Ambrose, 2001; Emery & Clayton, 2009).

In order to demonstrate that a task-aware latent space encodes useful affordance information we require a mechanism to train such a model as well as to purposefully explore the space. To this end we propose an architecture in which a task-based performance predictor (a classifier) operates on the latent space of a generative model (see fig. 1). During training the classifier is used to provide an auxiliary objective, aiding in shaping the latent space. Importantly, however, during test time the performance predictor is used to guide exploration of the latent space via activation maximisation (Erhan et al., 2009; Zeiler & Fergus, 2014; Simonyan et al., 2014), thus explicitly exploiting the structure of the space. While our desire to affect factors of influence is similar in spirit to the notion of disentanglement, it contrasts significantly with models such as $\beta$-VAE (Higgins et al., 2017), where the factors of influence are effectively encouraged to be axis-aligned. Our approach instead relies on a high-level auxiliary loss to discover the direction in latent space to explore.

Our experiments demonstrate that artificial agents are able to *imagine* an appropriate tool for a variety of reaching tasks by manipulating the tool's task-relevant affordances. To the best of our knowledge, this makes us the first to demonstrate an artificial agent's ability to imagine, or synthesise, 3D meshes of tools appropriate for a given task via optimisation in a structured latent embedding.

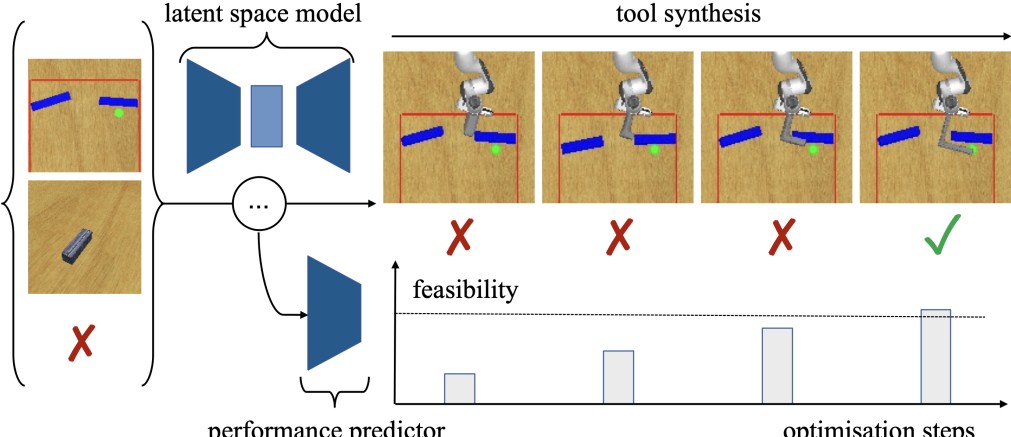

Figure 1: Tool synthesis for a reaching task. Our model is trained on data-triplets {task observation, tool observation, success indicator}. Within a scenario, the goal is to determine if a given tool can reach the goal (green) while avoiding barriers (blue) and remaining behind the boundary (red). If a tool cannot satisfy these constraints, our approach (via the performance predictor) *imagines* how one may augment it in order to solve the task. Our interest is in what these augmentations, imagined during *tool synthesis*, imply about the learned object representations.

Similarly, while activation maximisation has been used to visualise modified input images before (e.g. Mordvintsev et al., 2015), we believe this work to be the first to effect deliberate manipulation of factors of influence by chaining the outcome of a task predictor to the latent space, and then decoding the latent representation back into a 3D mesh. Beyond the application of tool synthesis, we believe our work to provide novel perspectives on affordance learning and disentanglement in demonstrating that object affordances can be viewed as *trajectories* in a structured latent space as well as by providing a novel architecture adept at deliberately manipulating interpretable factors of influence.

## 2 RELATED WORK

The concept of an *affordance*, which describes a potential action to be performed on an object (e.g. a doorknob *affords* being turned), goes back to Gibson (1977). Because of their importance in cognitive vision, affordances are extensively studied in computer vision and robotics. Commonly, affordances are learned in a supervised fashion where models discriminate between discrete affordance classes or predict masks for image regions which afford certain types of human interaction (e.g. Stoytchev, 2005; Kjellström et al., 2010; Tikhanoff et al., 2013; Mar et al., 2015; Myers et al., 2015; Do et al., 2018). Interestingly, most works in this domain learn from object shapes which have been given an affordance label a priori. However, the affordance of a shape is only properly defined in the context of a task. Hence, we employ a task-driven traversal of a latent space to optimise the shape of a tool by exploiting factors of variation which are conducive to task success.

Recent advances in 3D shape generation employ variational models (Girdhar et al., 2016; Wu et al., 2016) to capture complex manifolds of 3D objects. Besides their expressive capabilities, the latent spaces of such models also enable smooth interpolation between shapes. Remarkable results have been demonstrated including 'shape algebra' (Wu et al., 2016) and the preservation of object part semantics (Kohli et al., 2020) and fine-grained shape styles (Yifan et al., 2019) during interpolation. This shows the potential of disentangling meaningful factors of variation in the latent representation of 3D shapes. Inspired by this, we investigate whether these factors can be exposed in a task-driven way. In particular, we propose an architecture in which a generative model for 3D object reconstruction (Liu et al., 2019) is paired with activation maximisation (e.g. Erhan et al., 2009; Zeiler & Fergus, 2014; Simonyan et al., 2014) of a task-driven performance predictor. Guided by its loss signal, activation maximisation traverses the generative model's latent representations and drives an imagination process yielding a shape suitable for the task at hand.

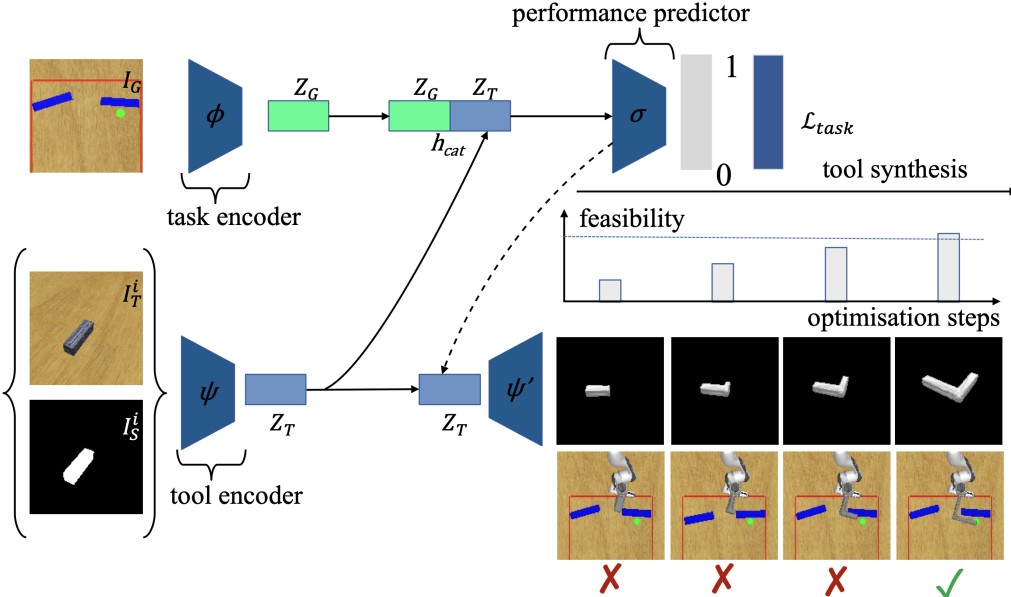

Figure 2: The model architecture. A convolutional encoder $\phi$ represents the task image $I_G$ as a latent vector $\mathbf{z}_G$. In parallel, the 3D tool encoder $\psi$ takes an input image $I_T^i$ and its silhouette $I_S^i$ and produces a latent representation $\mathbf{z}_T$. The concatenated task–tool representation $\mathbf{h}_{cat}$ is used by a classifier $\sigma$ to estimate the success of the tool at solving the task (i.e. reaching the goal). Given the gradient signal from this performance predictor for success, the latent tool representation $\mathbf{z}_T$ gets updated to render an increasingly suitable tool (via the 3D tool decoder $\psi'$). We pretrained the encoding and decoding models ($\psi$, $\psi'$) together as in prior work (Kato et al., 2018; Wang et al., 2018).

A key application of affordance-driven shape imagination is tool use. Robotics boasts a mature body of literature studying how robots can utilise tools to improve their performance across a wide range of tasks like reaching (Jamone et al., 2015), grasping (Takahashi et al., 2017), pushing (Stoytchev, 2005) and hammering (Fang et al., 2018). The pipeline executing tool-based tasks typically starts with models for tool recognition and selection (e.g. Tikhanoff et al., 2013; Zhu et al., 2015; Fang et al., 2018; Saito et al., 2018; Xie et al., 2019) before tool properties and affordances are leveraged to compute higher-order plans (Toussaint et al., 2018). Our proposed model lends itself to robotics applications like these, as the learned latent space encodes a rich object-centric representation of tools that are biased for specific tasks.

## 3 METHOD

Our overarching goal is to perform task-specific tool synthesis for 3D reaching tasks. We frame the challenge of tool imagination as an optimisation problem in a structured latent space obtained using a generative model. The optimisation is driven by a high-level, task-specific performance predictor, which assesses whether a target specified by a goal image $I_G$ is reachable given a particular tool and in the presence of obstacles. To map from tool images into manipulable 3D tools, we first train an off-the-shelf 3D single-view reconstruction model taking as input tool images $I_T^i, I_T^j$ and corresponding tool silhouettes $I_S^i, I_S^j$ as rendered from two different vantage points $i$ and $j$. After training, the encoder can infer the tool representation that contains the 3D structure information given a single-view RGB image and its silhouette as input. This representation is implicitly used to optimise the tool configuration to make it suitable for the task at hand. An overview of our model is given in fig. 2.

More formally, we consider $N$ data instances: $\{(I_G^n, I_T^{n,i}, I_T^{n,j}, I_S^{n,i}, I_S^{n,j}, \rho^n)\}_{n=1}^N$, where each example features a task image $I_G$, tool images $I_T$ in two randomly selected views $i$ and $j$, and their

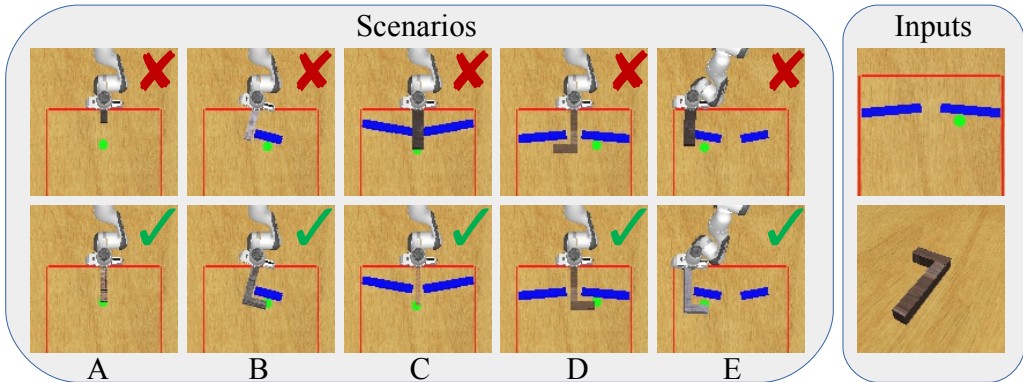

Figure 3: (Left) Task examples from our dataset. Top and bottom rows correspond to unsuccessful and successful tool examples respectively. Columns A - E represent five different task scenario types each imposing different tool constraints including width, length, orientation and shape. Note that the robot is fixed at its base on the table and constrained to remain outside the red boundary. Hence, it can only reach the green target with a tool while avoiding collisions with the blue obstacles. (Right) Model inputs {task observation, tool observation} during training and test time.

corresponding silhouettes $I_S$, as well as a binary label $\rho$ indicating the feasibility of reaching the target with the given tool. Examples of task images and model inputs are shown in fig. 3. In all our experiments, we restrict the training input to such sparse high-level instances. For additional details on the dataset, we refer the reader to the supplementary material.

### 3.1 REPRESENTING TASKS AND TOOLS

Given that our tools are presented in tool images $I_T$, it is necessary for the first processing step to perform a 3D reconstruction of $I_T$, from pixels into meshes. To achieve this single view 3D reconstruction of images into their meshes, we employ the same architecture as proposed by (Kato et al., 2018; Wang et al., 2018). The 3D reconstruction model consists of two parts: an encoder network and a mesh decoder. Given the tool image and its silhouette in view $i$, i.e $I_T^i$ and $I_S^i$, we denote the latent variable encoding the tool computed by the encoder, $\psi$, as

$$\psi(I_T^i, I_S^i) = \mathbf{z}_T. \tag{1}$$

The mesh decoder takes $\mathbf{z}_T$ as input and synthesises the mesh by deforming a template. A differentiable renderer (Liu et al., 2019) predicts the tool's silhoutte $\hat{I}_S^j$ in another view $j$, which is compared to the ground-truth silhouette $I_S^j$ to compute the silhouette loss $\mathcal{L}_s$. This silhouette loss $\mathcal{L}_s$ together with an auxiliary geometry loss $\mathcal{L}_g$ formulates the total 3D reconstruction loss:

$$\mathcal{L}_{recon} = \mathcal{L}_s + \mu \mathcal{L}_g, \tag{2}$$

where $\mu$ is the weight of the geometry loss. We refer the reader to Liu et al. (2019) regarding the exact hyper-parameter and training setup of the 3D reconstruction model.

Task images $I_G$ are similarly represented in an abstract latent space. For this we employ a task encoder, $\phi$, which consists of a stack of convolutional layers.[1] $\phi$ takes the task image $I_G$ as input and maps it into the task embedding $\mathbf{z}_G$.

### 3.2 TOOL IMAGINATION

**Task-driven learning**   The tool representation $\mathbf{z}_T$ contains task-relevant information such as tool length, width, and shape. In order to perform tool imagination, the sub-manifold of the latent space that corresponds to the task-relevant features needs to be accessed and traversed. This is achieved by adding a three-layer MLP as a classifier $\sigma$. The classifier $\sigma$ takes as input a concatenation $\mathbf{h}_{cat}$ of the task embedding $\mathbf{z}_G$ and the tool representation $\mathbf{z}_T$, and predicts the softmax over the binary task

---

[1]Architecture details are provided in the supplementary material.

success. The classifier learns to identify the task-relevant sub-manifold of the latent space by using the sparse success signal $\rho$ and optimising the binary-cross entropy loss, such that

$$\mathcal{L}_{task}\left(\sigma\left(\mathbf{h}_{cat}\right), \rho\right) = -\left(\rho\log\left(\sigma\left(\mathbf{h}_{cat}\right)\right) + \left(1-\rho\right)\log\left(1-\sigma\left(\mathbf{h}_{cat}\right)\right)\right), \tag{3}$$

where $\rho \in \{0,1\}$ is a binary signal indicating whether or not it is feasible to solve the task with the given tool. The whole system is trained end-to-end with a loss given by

$$\mathcal{L}\left(I_G, I_T^i, I_S^i, I_T^j, I_S^j, \rho\right) = \mathcal{L}_{recon} + \mathcal{L}_{task}. \tag{4}$$

Note that the gradient from the task classifier $\sigma$ propagates through both the task encoder $\phi$ and the toolkit encoder $\psi$, and therefore helps to shape the latent representations of the toolkit with respect to the requirements for task success.

**Tool imagination**  Once trained, our model can synthesise new tools by traversing the latent manifold of individual tools following the trajectories that maximise classification success given a tool image and its silhouette (fig. 2). To do this, we first pick a tool candidate and concatenate its representation $\mathbf{z}_T$ with the task embedding $\mathbf{z}_G$. This warm-starts the imagination process. The concatenated embedding $\mathbf{h}_{cat}$ is then fed into the performance predictor $\sigma$ to compute the gradient with respect to the tool embedding $\mathbf{z}_T$. We then use activation maximisation (Erhan et al., 2009; Zeiler & Fergus, 2014; Simonyan et al., 2014) to optimise $\mathbf{z}_T$ with regard to $\mathcal{L}_{task}$ of the success estimation $\sigma\left(\mathbf{h}_{cat}\right)$ and a feasibility target $\rho_s = 1$, such that

$$\mathbf{z}_T = \mathbf{z}_T + \eta\frac{\partial\mathcal{L}_{task}\left(\sigma\left(\mathbf{z}_T\right), \rho_s\right)}{\partial\mathbf{z}_T}, \tag{5}$$

where $\eta$ denotes the learning rate for the update. Finally, we apply this gradient update for $S$ steps or until the success estimation $\sigma\left(\mathbf{z}_T\right)$ reaches a threshold $\gamma$, and use $\psi'(\mathbf{z}_T)$ to generate the imagined 3D tool mesh represented by $\mathbf{z}_T$.

## 4 EXPERIMENTS

In this section we investigate our model's abilities in two experiments. First, we verify the functionality of the task performance predictor $\sigma$ in a *tool selection* experiment where only one out of three tools is successfully applicable. Second, we examine our core hypothesis about task-driven tool synthesis in a *tool imagination* experiment where the model has to modify a tool shape to be successfully applicable in a given task. In both experiments, we compare our full *task-driven* model, in which the tool latent space was trained jointly with the task performance predictor, with a *task-unaware* baseline, in which the 3D tool representation was trained first and the task performance predictor was fitted to the fixed tool latent space. We report our results in table 1 as mean success performances within a 95% confidence interval around the estimated mean.

**Tool Selection**  We verify that the classifier $\sigma$ correctly predicts whether or not a given tool can succeed at a chosen task. For each task, we create a toolkit containing three tool candidates where exactly one satisfies the scenario constraints. The toolkits are sampled in the same way as the remaining dataset and we refer the reader to fig. 3 again for illustrations of suitable and unsuitable tools. We check whether the classifier outputs the highest success probability for the suitable tool. Achieved accuracies for tool selection are reported in the left column of table 1.

**Tool Imagination**  We evaluate whether our model can generate tools to succeed in the reaching tasks. For each instance the target signal for feasibility is set to $\rho_s = 1$, i.e. *success*. Then, the latent vector of the tool is modified via backpropagation using a learning rate of 0.01 for 10, 000 steps or until $\sigma(\mathbf{h}_{cat})$ reaches the threshold of $\gamma = 0.997$. The imagined tool mesh is generated via the mesh decoder $\psi'$. This is then rendered into a top-down view and evaluated using a feasibility test which checks whether all geometric constraints are satisfied, i.e. successful reaching from behind the workspace boundary while not colliding with any obstacle. We report the percentage of imagined tools that successfully pass this test in table 1.

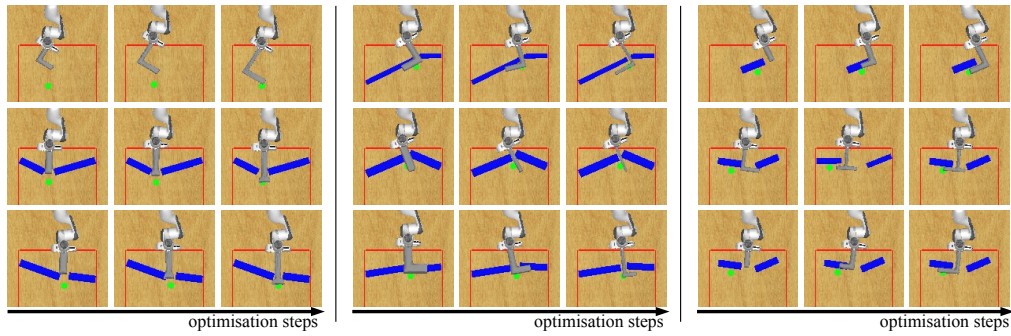

Figure 4: Qualitative results of tool evolution during the imagination process. Each row illustrates an example of how the imagination procedure can succeed at constructing tools that solve the task by: (left) increasing tool length, (middle) decreasing tool width, and (right) altering tool shape (creating an appropriately oriented hook). Each row in each grid represents a different imagination experiment.

### 4.1 MODEL TRAINING

In order to gauge the influence of the task feasibility signal on the latent space of the tools, we train the model in two different setups. A *task-driven* model is trained with a curriculum: First, the 3D reconstruction module is trained on tool images alone. Then, the performance predictor is trained jointly with this backbone, i.e. the gradient from the predictor is allowed to back-propagate into the encoder of the 3D reconstruction network. In a *task-unaware* ablation, we keep the pre-trained 3D reconstruction weights fixed during the predictor training removing any influence of the task performance on the latent space. All models are trained for 15,000 steps in total. The first 10,000 steps are spent on pre-training the 3D reconstruction part in isolation and the remaining 5,000 steps are spent training the task performance predictor. We select checkpoints that have the lowest task loss $\mathcal{L}_{task}$ on the validation split.

### 4.2 QUANTITATIVE RESULTS

We start our evaluation by examining the task performance predictor in the tool selection experiment (TS). For each scenario type (A - E) we present 250 tasks from the test set, each paired with three tool images and their respective silhouettes. We encode each tool with the tool encoder $\psi$ and concatenate its representation $\mathbf{z}_T$ to $\mathbf{z}_G$ obtained from the task image encoder $\phi$. Each concatenated pair $h_{cat}$ is passed through the feasibility predictor $\sigma$ and the tool which scores highest is selected. We report the results for this experiment in the left section of table 1. The results confirm that $\sigma$ is able to identify suitable tools with almost perfect accuracy. Tool selection accuracy does not differ significantly between the task-driven and the task-unaware variations of the model. This suggests

Table 1: (Left) Tool Selection: Mean accuracy when predicting most useful tool among three possible tools. (Right) Tool Imagination: Comparison of imagination processes when artificially warm-starting from the same unsuitable tools in each instance. Best results are highlighted in bold.

| Scn | N | Tool Selection Success [%] | | Tool Imagination Success [%] | | |
|-----|-----|------------|------------|-------------|--------------|-------------|
| | | Task-Unaware | Task-Driven | Random Walk | Task-Unaware | Task-Driven |
| A | 250 | $88.8 \pm 3.9$ | $90.8 \pm 3.6$ | $3.6 \pm 2.3$ | $55.6 \pm 6.2$ | $\mathbf{96.4 \pm 2.3}$ |
| B | 250 | $96.4 \pm 2.3$ | $97.6 \pm 1.9$ | $5.6 \pm 2.9$ | $42.0 \pm 6.1$ | $\mathbf{78.8 \pm 5.1}$ |
| C | 250 | $96.4 \pm 2.3$ | $97.2 \pm 2.1$ | $23.6 \pm 5.3$ | $56.8 \pm 6.1$ | $\mathbf{76.4 \pm 5.3}$ |
| D | 250 | $96.8 \pm 2.2$ | $98.4 \pm 1.6$ | $2.4 \pm 1.9$ | $\mathbf{75.2 \pm 5.4}$ | $81.2 \pm 4.8$ |
| E | 250 | $87.2 \pm 4.1$ | $87.6 \pm 4.1$ | $13.6 \pm 4.3$ | $\mathbf{88.4 \pm 4.0}$ | $86.4 \pm 4.3$ |
| Tot | 1250 | $93.1 \pm 1.4$ | $94.3 \pm 1.3$ | $9.8 \pm 1.7$ | $63.6 \pm 2.7$ | $\mathbf{83.8 \pm 2.0}$ |

that the factors of tool shape variation are captured successfully in both cases and the feature vectors produced by $\psi$ are discriminative enough to be separated efficiently via the MLP $\sigma$.

After verifying the task performance predictor's functionality in the tool selection experiment, we investigate its ability to drive a generative process in the next experiment. This is done to test our hypothesis about the nature and exploitability of the latent space. Given that the latent space captures factors of variation in 3D tool geometry, we hypothesise that these factors can be actively leveraged to synthesise a new tool by focusing on the performance predictor. Specifically, we present our model with a task image $I_G$ and an unsuitable tool geometry. We then encode the tool via $\psi$ and modify its latent representation $\mathbf{z}_T$ by performing activation maximisation through the performance predictor $\sigma$. As the feasibility prediction is pushed towards 1, the tool geometry gradually evolves into one that is applicable to the given task image $I_G$. In addition to comparing the imagination outcomes for the task-driven and the task-unaware model, we also include a *random walk* baseline, where, in place of taking steps in the direction of the steepest gradient, we move in a *random direction* in the task-driven latent space for $10,000$ steps. In this baseline the latent vector of the selected tool is updated by a sample drawn from an isotropic Gaussian with mean 0, and, to match the step size of our approach, the absolute value of the ground-truth gradient derived by back-propagating from the predictor as the variance.

For 250 instances per scenario type, we warmstart each imagination attempt with the same infeasible tool across random walk, task-driven, and task-unaware models to enable a like-for-like comparison, with the results presented in table 1. The performance of the random walk baseline reveals that a simple stochastic exploration of the latent space is not sufficient to find suitable tool geometries. However, following the gradient derived from the performance predictor leads to successful shaping of tool geometries in a much more reliable way. While the task-unaware ablation provides a strong baseline, transforming tools successfully in 63.6% of the cases, the task-driven model significantly outperforms it, achieving a global success rate of 83.8% on the test cases. This implies that jointly training the 3D latent representation and task performance predictor significantly shapes the latent space in a 'task-aware' way, encoding properties which are conducive to task success (e.g. length, width, and configuration of a tool) along smooth trajectories. Moreover, each of these trajectories leads to higher *reachability* suggesting that these affordances can be seen as a set trajectories in a task-aware latent space.

### 4.3 QUALITATIVE RESULTS

Qualitative examples of the tool imagination process are provided in fig. 4 and fig. 5. In the right-middle example of fig. 4, a novel T-shape tool is created, suggesting that the model encodes the vertical stick-part and horizontal hook-part as distinct elements. The model also learns to interpolate the direction of the hook part between pointing left and right, which leads to a novel tool. As shown in fig. 5, tools are modified in a smooth manner, leading us to hypothesise that tools are embedded in a continuous manifold of changing length, width and configuration. Optimising the latent embedding for the highest performance predictor score often drives the tools to evolve along these properties. This suggests that these geometric variables are encoded as *trajectories* in the structured latent space learnt by our model and deliberately traversed via a high-level task objective in the form of the performance predictor.

## 5 CONCLUSION

In this paper we investigated the ability of an agent to synthesise tools via task-driven imagination within a set of simulated reaching tasks. Our approach explores a hybrid architecture in which a high-level performance predictor drives an optimisation process in a structured latent space. The resulting model successfully generates tools for unseen scenario types not in the training regime; it also learns to modify interpretable properties of tools such as length, width, and shape. Our experimental results suggest that these object affordances are encoded as *trajectories* in a learnt latent space, which we can navigate through in a deliberate way using a task predictor and activation maximisation, and interpret by decoding the updated latent representations. Ultimately, this may aid in our understanding of object affordances while offering a novel way to disentangle interpretable factors of variation – not only for 3D tool synthesis. To facilitate further work in this area, we plan to release both the reaching dataset and trained model to the community.

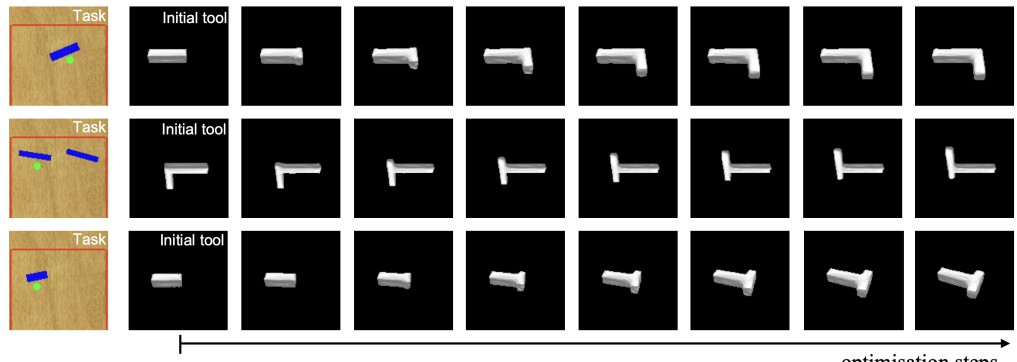

optimisation steps

Figure 5: Examples of tool synthesis progression during the imagination process. In the top row, a stick tool morphs into a hook. The middle row shows a left-facing hook transforming into a right-facing hook. In the bottom row, the tool changes into a novel T-shape. Constraints on these optimisations are specified via task embeddings corresponding to the task images on the far left.

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
