# OpenReview forum: "Imagine That! Leveraging Emergent Affordances for 3D Tool Synthesis"
_ICLR.cc/2021/Conference — Reject_

### Official Review · AnonReviewer4 · 2020-10-27
**Initial review**

**Rating:** 5
**Confidence:** 3

**Review:**

Summary
-------
By combining a task-success classifier with the latent space of a tool-shape generative model, this paper shows that an activation-maximization approach can generate tool shapes which can succeed at particular tasks.

Positives
---------
The paper addresses the interesting topic of affordances, and how latent representations of tool shapes might be structured.

The methods and experiments are clean and well motivated, and the writing is clear.

Negatives
---------
The specific contribution of the paper is difficult to see at first glance.  The architecture is taken from previous work, as well as the method for maximizing activations.

Although the task-unaware baseline is a reasonable one to compare against, I would like to see other comparisons to approaches present in the literature.  For example, how do the supervised affordance learning approaches perform in the tool imagination task when combined with task-success prediction?  Additionally, are other methods for latent disentangling sufficent to observe similar behavior?

In the tool imagination task, it would be useful to see a couple of things from the task-unaware approach to make sure the conclusions are sound.  First, how well does the activation-maximization step work?  Is the model reaching the same level of feasibility for task aware and unaware versions?  Second, what do the imagined tool trajectories look like for the task unaware approach?  If they are also smooth and encode length, width, etc., it would be hard to claim that task-aware approaches are required for that type of encoding.

Reasons for score
-----------------
The topic explored is interesting, and the experiments are simple and illustrative, but there are remaining questions about the baseline comparisons required to make strong conclusions.

Post-rebuttal response
-----------
Thank you for the detailed rebuttal responses.  The rebuttal suggests that there are not many other baselines against which to compare.  If that is true, I would want to see much more detailed analysis of the comparisons offered in the paper.  However, I am still missing details of the latent space generated by the task-unaware approach.  I will leave my score as it is.

---

### Official Review · AnonReviewer3 · 2020-10-28

**Rating:** 4
**Confidence:** 5

**Review:**

The authors try to tackle the problem of tool synthesis by using a classifier to guide the learning and exploitation of a generative model through activation maximization. Experiments are conducted on the proposed simulated reaching dataset on tool selection and tool imagination.  While the idea of synthesizing tools step by step continuously is interesting, the technical and experimental design make the problem over-simplified and thus raise some concerns.

The overall concern for the paper is two-fold: 1) Some overclaims. Affordance is often used to describe the potential action that can be applied to certain objects. In this case only the shape of tool is changing however the concept of `"emergent affordances" is hardly touched. Also current proposed dataset only tackle reaching task in a static way. So the problem degenerates to a geometric constraint satisfaction problem. It would be interesting to add more different tasks and potentially extend the current binary label for task success.  2) The experiment design is weak.  For the tool selection task, the task-unaware and task-driven model achieve similar result, which means information from task predictor is not helpful in this case. In tool imagination task, the gradient doesn't flow back to the learned representation in the task-unaware case so suboptimal performance understandable. It would be better if the test setting is more diversified with more degree of freedom in the simulation dataset to showcase the capability of proposed model.

The paper is easy to follow despite some typos, e.g. the bold numbers in Table1.

Some detail questions:
What are the differences for 5 types of the scenarios in detail?
How many images of tools are used to train 3D reconstruction model?
Is the position/orientation of the tool fixed in the tool selection and tool imagination evaluation? Is it possible that a tool that cannot finish the reaching task at one location can reach the target when positioned at another location/direction?
Are there any failure cases in tool imagination and what might be the reason?
In tool selection task, how are the tools sampled apart from the ground truth?

------
Post rebuttal:
After reading the author's response as well as the opinions from other reviewers, I will stick to my original rating. Although the authors resolve some of the concerns in the rebuttal, there are still limitations in the method and task design.

---

### Official Review · AnonReviewer1 · 2020-10-29
**Is imagination necessary here?**

**Rating:** 4
**Confidence:** 3

**Review:**

This paper presents a method with two goals: (1) estimate if a given tool can solve a given task, and (2) generate a tool that can solve a given task. One encoder maps a tool image and silhouette into a latent code, and a decoder maps this code into a mesh; another encoder maps a task image into a latent code; this code is concatenated with the tool code, then mapped to a task success probability. The networks are trained together, so that task success probability has some impact on the tool encoder's latent space. Results show that the model is largely successful in both tasks, but no challenging baselines are here.

I think the idea overall is creative and well-executed, and the paper is well-written.

I was happy to see that the supplemental includes information about how feasibility labels are generated. I think this should be in the main paper.

What is the geometry loss L_g? I looked in the cited paper and saw that details are bare there too -- it only says "regularizes the Laplacian of both shape and color predictions". What does this mean exactly, and why does it help?

Why is the success signal \rho called "sparse" (top of page 5)? My understanding is that this label is available on every training example.

When I think of "imagination", I think of interpolating or extrapolating into space that was not quite seen at training time. This leads to two ideas:

1. The task is simple enough that imagination may not be necessary. How about a nearest neighbors baseline? Classify the task using nearest neighbor image retrieval (searching in the training set, using maybe the task encoding z_G), and use the tool that was successful in that training example.

2. How about creating a setup where imagination is definitely necessary? e.g., generate a scenario that requires a tool in-between one of the tools seen at training time. Maybe train with long sticks and short L's, and create a task that demands a long L. This would show that the model has a useful latent space and that the imagination (i.e., latent space traversal) is beneficial. The paper notes a novel "T-shape tool" but I am not sure this counts, since in Figure 4 the T is only some intermediate stage of the optimization, and in Figure 5, a T is a strange choice for the task shown (in row 3), so it seems like traversal simply stopped when the horizontal part shifted far enough, and there is no purpose to the "T" tool per se.


-------------
Post-rebuttal: After reading the rebuttal and the other reviews, I have decided to lower my rating. The rebuttal appears to state that all baselines and evaluations suggested by the reviewers are irrelevant to the goals of the paper. I see the two main hypotheses clarified in the rebuttal, but I do not see a convincing argument for avoiding comparisons to related approaches to the task at hand.

---

### Official Review · AnonReviewer2 · 2020-10-29
**Interesting problem, but some key ingredients are missing. Experimental results are insufficient.**

**Rating:** 4
**Confidence:** 5

**Review:**

This paper explores the idea of tool synthesis in an unsupervised generative learning setting.

Pros:

+ Tool use is one of the crucial aspects to demonstrate human-like intelligence. The authors' proposal to synthesize the tool's shapes for a given task is a fundamental research problem. This is different from most prior work that focuses on ranking/selection or simply categorization. Tool synthesis is particularly difficult; only one paper was published recently on this particular topic [1]. It has not yet passed the peer-review, but I recommend the authors consider citing it for future readers for completeness purposes.

+ Different from classic object recognition, the tool use is intrinsically a task-oriented concept. This paper did a few things right: emphasis on the task-oriented view, no affordance label, etc. The authors are definitely heading in the right direction.

Cons:

- What is the representation of the task or the tool? This is probably the most significant difference that could tell the present work from the prior work, especially if the results are promising. However, the authors only spend three short paragraphs in the paper (section 3.1), which does not really touch this point, despite the section title is called "representing tasks and tools." It seems to be a simple, off-the-shelf encoder that has been published. Representing tools is significant as making/synthesizing tools requires a deep understanding of physics and causality (for instance, see the cited Zhu et al 2015 and Toussint et al 2018). Without a proper representation of these concepts or a clear disentanglement of the model, one simply cannot justify their model is learning how to synthesize tools. Instead, the model is more likely to learn an association between the given task and a trained shape (or 3D shape space if learned better). In this view, it really does not touch the core of the computational problem of tool-use: The problem of tool synthesis becomes 3D shape synthesis without really understanding why an object becomes a tool in that context, and the crucial concept of "task-driven" becomes a simple label.

- What about action? A tool-use or task-oriented view cannot be separated from the actual action. This is one of the key ideas behind affordance in Gibson's theory. Given different embodiments of the agent with various capabilities of actions, the synthesized tool would be dramatically different. This is a very challenging problem in tool manipulation in robotics (e.g., see [2]).

- Furthermore, what about trajectories? Even if an algorithm can synthesize the tool and chooses an action, one still needs to properly manipulate tools with the planned trajectory to complete the task. This seems to be completely left out in the paper.

- Why not directly use 3D meshes as input for the algorithm, instead of using two views to reconstruct? The reconstructed one does not bring in any benefit for the overall goal of the paper.

- The experiments are far too simple. The authors do have extra space on page 8, but did not include additional results. This makes me wonder how robust the algorithm is.

[1] Tool MacGyvering: A Novel Framework for Combining Tool Substitution and Construction
[2] Mirroring without Overimitation: Learning Functionally Equivalent Manipulation Actions

---

### Author Response · Authors · 2020-11-16
**Rebuttal Part I**

We thank the reviewers for their time, effort and  constructive comments and are delighted about the general enthusiasm the reviews exhibit for the idea we are pursuing. The principal criticisms offered by the reviews refer to (i) overly simplistic experimental setups and (ii) a lack of baseline comparisons. We will tackle these as well as additional points made by individual reviewers below.

Our submission investigates two succinct and intimately related hypotheses: (a) object affordances are implicitly captured in structured latent spaces trained based on experience (task success) alone; and (b) they are specifically addressable in a similarly task-driven way. We are not aware of any other works either postulating these hypotheses or offering any experimental insight into whether they have merit. And we interpret all reviews to agree that this direction of investigation is unusual, novel and interesting.

- Overly simplistic experimental setup; Degenerates to geometric constraint satisfaction.  [AR2, AR3]
Our experiments are designed to confirm or reject our research hypotheses. Rather than addressing all possible object affordances we select a subset relating to object geometry: length, width and configuration. This is done precisely because it lends itself to simple, easily reproducible experimental setups. The “degeneration” to geometric constraint satisfaction, therefore, is a deliberate design decision to allow for a clear and targeted proof of concept. There is, however, nothing in our architecture that limits our approach to these particular affordances.

- Tool synthesis requires an understanding of physics and causality [AR2].
It is tempting to agree with this. However, the comment suggests that meaningful progress in representing objects and their affordances can only be made once understanding physics and causality have been solved. We do not believe that this is the case - and we offer our submission as evidence to the contrary. Not all scenarios requiring tool synthesis require physical and/or causal understanding. The foundational interest in similar reaching tasks in the experimental psychology literature is a case in point that such a task simplification can be instructive when it comes to understanding spatial reasoning - not despite the simplified setup, but because of it.

- What about actions? What about trajectories? [AR2]
Agents and their actions are intricately interwoven with the notion of object affordances. However, in order to disambiguate the role of various agents (which rightly introduce additional complexity) from the effect of task success or failure on object representations we control for the set of agents by choosing only a single one, which is able to act as required to manipulate the objects considered. We agree that this is a significant simplification of the problem space. And we argue that this is exactly what is required in the context of our research hypotheses. By considering agent experience (via task success) our framework seamlessly extends to cases of different agents. However, in the context of our submission we believe such an investigation distracts from the specific hypotheses we have set out to investigate.

- Task-driven becomes a simple label [AR2].
Correct. This is, in fact, one of the merits of our approach. Task success is an easily quantifiable measure. It is ubiquitous in the robotics literature. The fact that we observe meaningful changes of tool configuration based simply on a representation learned via the high-level signal of task success is, we believe, not just interesting but noteworthy.

- No challenging baselines; What about other methods of latent disentanglement?
 [AR1, AR4]
We thank the reviewers for the suggestions regarding baselines: (a) nearest neighbour classification of successful tools; (b) supervised affordance learning combined with success predictors; and (c) other methods of latent disentanglement. However, we posit that (a) and (b) do not add value to our investigation into the ability of structured representations learned with minimal supervision to implicitly encode specific, task-relevant object affordances. In this context both (a) and (b) simply amount to supervised tool selection, which is interesting but largely orthogonal to our research direction. In our work we employ tool selection only as a means of validating parts of our architecture, which is otherwise designed to effect tool synthesis. (c) is an interesting proposition and would add value if our principal claim was that our way of achieving disentanglement was superior to those already proposed in the literature. We do not claim this. Instead we propose an architecture which naturally accomplishes disentanglement in the same task-driven framework used for representation learning.

---

### Author Response · Authors · 2020-11-16
**Rebuttal Part II**

As noted by several reviewers, our direction of travel is interesting and creative. It is novel in the context of representation learning for object affordances. We are unaware of any other works exploring similar research hypotheses. Consequently, we were unable to identify suitable baselines.

- What are the differences between the five scenario types? Can a tool that does not succeed in one place succeed in another? Generate scenarios where tools in-between are required? [AR1, AR3]
Beyond tool synthesis, the agent is free to choose tool position and orientation for all scenarios - so yes, a tool that does not succeed in one place can succeed in another. The only constraints are that the agent itself is not allowed to enter the red work space. The scenarios are designed such that they encourage the emergence of different aspects of tool synthesis. [A] is entirely unconstrained (position, length), [B] does not allow direct access to the target but is otherwise unconstrained (length and tool geometry), [C] requires a bottleneck to be traversed before accessing the target (encourages a change in length and width), [D] contains a target off-centre beyond a bottleneck (so encourages a change length, width and tool geometry), [E] provides multiple bottlenecks (at this point all aspects investigated are required to vary). One might argue that any tool synthesised here is an “in-between” tool in terms of the affordances investigated. What we do not show here explicitly is tool extrapolation (e.g. to shapes not previously seen). This is on our roadmap for future work.

---

### Decision · Program_Chairs · 2021-01-07
**Final Decision**

**Decision:**

Reject

**Comment:**

This paper proposes a method for tool synthesis by jointly training a generative model over meshes and a task success predictor. Gradient-based planning is then used to find a latent space tool representation which maximizes task success, given a starting tool and an input scene. The results indicate that this method can successfully generate simple tools, and that it performs better than either a random baseline or a version where the generative model and success predictor are trained independently.

The reviewers unanimously felt that this paper was not quite ready for publication at ICLR. While I'm a strong believer that unique and creative papers which tackle understudied problems---such as this one---ought to be encouraged, and that the authors' rebuttal satisfactorily addressed most of the reviewers' concerns, there was one major point that was not. In particular, all reviewers noted that the paper lacks comparison to convincing baselines and/or sufficiently extensive experiments. While I do not think baselines are necessary per se, especially in such a unconventional setting such as this, I believe what the reviewers are getting at (and I agree) is that the results as presented don't really help the reader understand the contours of the method and/or problem space, and as a result, the contributions of the paper feel thin. For example, here are some questions that the reviewers raised, which I do not feel were adequately addressed:

- R3: What are the failure cases of the model?
- R2: How important is the particular representation of the task and tool (i.e., visual for the task, meshes for the tool)?
- R4: How do the imagined tool trajectories compare between the task-aware and task-unaware cases?
- R4: Is the success classifier trained to the same level of performance in both task-aware and task-unaware settings? (In general, it would be helpful to include learning curves in the appendix.)
- R1: How important is the choice of the particular planning/optimization method (i.e. gradient descent)?
- R1: What is the generalization performance of the model along affordance directions (e.g. needing to synthesize longer/shorter tools than seen during training)?

Taken individually, such questions might not be an issue, but together they illustrate a larger concern that the paper has not done a thorough enough job of analyzing and evaluating the proposed method. Therefore, at this stage I recommend rejection. I think that by fleshing the paper out with some answers to the above questions, this could make an excellent submission to a future conference.